# Comparison of Operative Times in Primary Bilateral Total Knee Arthroplasty Performed by a Single Surgeon

**DOI:** 10.3390/jcm11164867

**Published:** 2022-08-19

**Authors:** Yoshinori Ishii, Hideo Noguchi, Junko Sato, Ikuko Takahashi, Hana Ishii, Ryo Ishii, Kei Ishii, Shin-ichi Toyabe

**Affiliations:** 1Ishii Orthopaedic & Rehabilitation Clinic, 1089 Shimo-Oshi, Gyoda, Saitama 361-0037, Japan; 2School of Plastic Surgery, Kanazawa Medical University, 1-1 Daigaku Uchinada, Ishikawa 920-0253, Japan; 3School of Orthopaedic Surgery, Shinshu University Hospital, 3-1-1 Asahi Matsumoto, Nagano 390-8621, Japan; 4Iwate Prefectural Ninohe Hospital, 38 Horino, Ninohe, Iwate 028-6193, Japan; 5Niigata University Crisis Management Office, Niigata University Hospital, Niigata University Graduate School of Medical and Dental Sciences, 1 Asahimachi Dori Niigata, Niigata 951-8520, Japan

**Keywords:** prolongation of operative time, bilateral primary TKA, body weight, calendar year of surgery, operative side, dominant side

## Abstract

Purpose: Prolonged operative time (OT) is associated with adverse complications after total knee arthroplasty (TKA). The purpose of this study was to determine whether preoperative factors, such as sex, age, body mass index, body weight (BW), body height, American Society of Anesthesiologists grade, tibiofemoral angle, hospital for special surgery scores, surgical side, surgical order, and calendar year of surgery, affect OT. Methods: One hundred and nineteen patients (238 knees) with osteoarthritis who underwent staged bilateral primary TKA performed by a single surgeon were evaluated. The medical records of 15 males and 104 females were retrospectively reviewed. All variables were expressed as median (interquartile range). Results: The OT for all TKAs was 57 min (51, 65). The OT on the left side (59 min (52–67)) was longer than that on the right side (55 min (50–62)) (*p* = 0.015). Multiple regression analysis revealed that longer OT was related to BW (β = 0.488, *p* < 0.001), calendar year of surgery (β = −0.218, *p*< 0.001), and operative side (β = −0.151, *p* = 0.007). The Jonckheere–Terpstra test showed a trend toward decreasing OT with calendar year on the left side (*p* = 0.037) (surgeon’s non-dominant side), but not on the right (*p* = 0.795). Body height, BW, and body mass index showed weak correlations (r = 0.212, *p* = 0.001; r = 0.352, *p* < 0.001; r = 0.290, *p* < 0.001, respectively) with OT. Conclusion: Patients with a large physique, and especially obesity, with an affected knee on the surgeon’s non-dominant side may require a longer OT; OT decreased over time.

## 1. Introduction

Prolonged operative time is associated with adverse complications after total knee arthroplasty (TKA), such as deep surgical site infection [1,2,3], venous thromboembolism [4], neurological deficits [5], and increased 30-day readmission [6] and mortality [7]. Factors related to prolonged operative time can be divided into two factors: health-care-provider-related and patient-related factors. Factors on the part of the health care provider include the surgeon’s and/or hospital’s low skill level [1,3], implant design [8], the lack of tourniquet [9], the use of patient-specific instruments [10], computer navigation [2], and mini-skin incisions [11,12]. Recently, Kaidi et al. [13] reported that intraoperative scrub nurse handoffs were associated with increased operative times for total joint arthroplasty. Patient-related factors reported to influence operative time include younger age [1], male [1,14,15], heavier body weight (BW) or higher body mass index (BMI) [3,16,17,18], American Society of Anesthesiologists (ASA) class [19] ≥ 3 [1], preoperative tibiofemoral angle (TFA) [17], previous surgery on the same knee [1], and a diagnosis of non-osteoarthritis (OA) [1]. However, to date, most reports except those of Kosashvili et al. [15] and Ishii et al. [17] have involved different surgeons and different patients. As a result, technical or anatomical bias, which seriously impacts operative time, is not excluded.

Therefore, the purpose of this study was to identify the factors affecting operative time by analyzing staged bilateral TKA patients, which is considered to involve less anatomical bias compared with analyses involving concurrent right and left knee surgeries. Additionally, the surgeries in this study were performed by a single surgeon, which might involve less technical bias than studies involving different surgeons. This study may contribute to devising strategies to further decrease operative time for primary TKA, from a more in-depth perspective than that previously reported.

## 2. Materials and Methods

Informed consent was obtained from all patients. The study received institutional review board approval. The analysis of pertinent data from patients who underwent primary TKA with the New Jersey low contact stress (LCS^®^) total knee system (DePuy, Warsaw, IN, USA) from February 1998 to October 2021 was performed. Overall, 119 patients (238 TKAs) who underwent staged bilateral TKA were included in this study. All surgeries were performed by one senior surgeon. 

The inclusion criterion was a diagnosis of primary osteoarthritis. Exclusion criteria comprised patients receiving different TKA designs on the left and right sides, previous revision arthroplasty or tibial osteotomy, or a history of rheumatoid arthritis. The side of the first TKA was chosen by the patient. The timing of the second TKA was determined solely by the patients according to their perceived ability to tolerate the additional pain and limitations to activities of daily living. 

Operative time was defined as the time from skin incision to wound closure. The following preoperative factors were analyzed: sex, ASA grade, age, BMI, BW, and TFA, which have previously been reported to affect operative time, and other factors, such as body height (BH), hospital for special surgery (HSS) scores [20], side of surgery, order (1st or 2nd) of surgeries, and calendar year of surgery.

### 2.1. Surgical Procedure

A single surgeon performed all procedures using a standardized technique with a tourniquet, which has been described in detail previously [21]. Two specific scrub nurses attended all TKAs without intraoperative nurse turnover. In all knees, the femoral components were fixed without cement; however, the tibial components were initially uncemented (from 1998 to 2002). After 2002, cement was used to fix the tibial components. Thus, 13 knees were cementless, and another 225 knees were fixed using a hybrid (cement only on the tibial side) technique. Only one patient underwent different tibial fixation (right side cement, left side cementless). No patellae were replaced. Proper intraoperative anteroposterior and abduction/adduction stability were confirmed manually, although these were not quantified intraoperatively. After TKA, the knees in all patients could achieve full extension and at least 90° of flexion intraoperatively while in the supine position. The clinical characteristics of the patients are summarized in Table 1. 

### 2.2. Statistical Analysis

Because the median interval between the first and second TKA surgeries was 628 days, the surgeries were treated as independent events. Comparisons between the first and second surgeries were evaluated without correspondence tests. Because some variables did not show a normal distribution with the Kolmogorov–Smirnov and Shapiro–Wilk’s test or the Q-Q plot, these variables were analyzed using a nonparametric method. All variables were expressed as median (25th percentile, 75th percentile).

The factors associated with operative time were examined by multiple regression analysis. The candidate independent variables entered into the multiple regression analysis were sex, age, BH, BW, BMI, HSS, TFA, calendar year of surgery, ASA grade, affected side, and the order of the surgeries regarding the side of the body. In a preliminary study, multicollinearity among these independent variables using the variable inflation factor (VIF) was checked. The results showed that BW, BH, and BMI had a VIF greater than 10. The VIF was less than 1 when BH and BMI were excluded; therefore, BH and BMI were not included in the multiple regression analysis in this study. A stepwise variable selection method was used to select the variables that were related to operative time.

Spearman’s rank sum test was used for correlation analysis between two variables, and a single regression analysis was used to examine the relationship between two variables. The difference between the two groups was assessed by the Wilcoxon rank sum test. The Jonckheere–Terpstra test was used to evaluate operative time trends by calendar year. The post hoc power of the Jonckheere–Terpstra test was also calculated.

All analyses were performed using IBM SPSS Statistics ver. 23 (IBM Japan, Tokyo, Japan) and R version 4.1.1 (GNU) with the related packages. In all tests, a *p*-value < 0.05 was considered statistically significant. 

## 3. Results

The median operative time for all 238 TKAs was 57 (min) (51, 65) (range, 39–116). When comparing right and left sides, the operative time on the right side was 55 (50, 62) (range, 40–104), and the time on the left side was 59 (52, 67) (range, 39–116). The operative time was significantly longer on the left side compared with the right side (*p* = 0.015) (Figure 1). Multiple regression analysis using the stepwise variable selection method revealed that longer operative time was related to BW, calendar year of surgery, and operative side (Table 2).

There was a significant association between operative time and calendar year in the multiple regression analysis, but there was no significant trend using the Jonckheere–Terpstra test (*p* = 0.134). The power of the one-sided Jonckheere–Terpstra test with an alpha error of 0.05 was examined. The post hoc power analysis of the test revealed a statistical power of 0.148, which was low and which may have prevented revealing a trend.

Because the surgical side was also related to operative time in the multiple regression analysis, the patients were divided into two groups according to the surgical side and we examined the trend in operative time with calendar year in each group. The results of the Jonckheere–Terpstra test showed a significant trend in decreasing operative time with calendar year for surgeries on the left side (*p* = 0.037) (Figure 2A) but not the right side (*p* = 0.795) (Figure 2B).

The correlations between operative time and the variables are shown in Table 3. BH, BW (Figure 3), and BMI had weak correlations (r = 0.212, *p*= 0.001; r = 0.352, *p* < 0.001; and r = 0.290, *p* < 0.001, respectively) with operative time (Table 3). However, the other variables, namely sex, age, HSS score, TFA, and order of surgery showed no correlations.

## 4. Discussion

Our results revealed three important findings. First, three factors, BW, calendar year of TKA surgery, and operative side, showed correlations with operative time in the multiple regression analysis. Second, operative side and BH, BW, and BMI showed correlations with operative time in the univariate analysis. Third, regarding the surgical side, operative time showed a correlation with the calendar year only with left-sided surgery, according to the trend test.

Patient BW was most associated with longer operative time, which has been reported in previous studies [16,17,18]. It is possible that surgery for patients with a heavy BW requires a longer time for exposing and closing the operative field, which may be associated with thickening of soft tissues, such as adipose tissue and muscle. Heavy BW might also induce an increase in bone mineral density in the femur and tibia [22], resulting in prolonging the time required for osteotomy for the preparation of component insertion [3]. Considering these conditions, it is reasonable that there was a correlation between larger patient physique and longer operative time in the univariate analysis.

The effects of the learning curve on decreasing operative time after accumulated experience performing TKA surgeries have been demonstrated, especially when introducing new techniques in TKA, such as a mini-skin incision [11,12], patient-specific instrumentation [10], computer navigation [1], and robotic-assisted techniques [23]. Similar to these previous reports, advancing calendar year, which may be associated with learning curve effects, showed a correlation with operative time in the multiple regression analysis. It was speculated that the following reasons explained this effect: The surgeon in this study already had enough experience performing TKA at the beginning of this study as reported [24,25], and might have above or equivalent skill to the standard. Thus, the length of less than 2 years, which was the median interval between TKA in each knee, appears to be insufficient to lead to remarkable changes in the surgeon’s surgical skills to the extent that the operative time was shortened. In addition, Naranje et al. [3] reported that the effect of greater surgical volume was surprisingly modest (approximately 5 min’ shorter operative time) and appeared to plateau after 300 TKAs. Therefore, greater volume did not decrease operative time. Considering Naranje et al.’s findings, the surgeon’s skill in this study might have plateaued because he had already performed over 300 TKAs 7 years ago [26]. Therefore, it seems rational that the main reason explaining the shorter operative time over time was not increased surgeon’s skill but rather efficient instrumentation handoffs or delivery between the surgeon and the scrub nurses.

The current study showed laterality regarding operative time between the left and right sides. It was speculated that the main reason was a “distance difference” between the surgeon and scrub nurse during TKA. Further distances between the surgeon and the scrub nurse may lead to longer times to perform instrument handoffs or deliveries. During TKA, the surgeon usually stands on the side of the affected knee, and scrub nurses usually stand at the caudal (foot) side of the patient lying in the supine position. Therefore, scrub nurses are further from the surgeon’s dominant hand during non-dominant side surgery compared with the distance during dominant side surgery. In this study, the distance between the surgeon and scrub nurse was greater during left-side TKA than during right-side TKA. Because the number of procedural steps in TKA between the surgeon and nurse range from 159 to 230 [27], even if the time discrepancy for one step between dominant- and non-dominant-side TKA might be only 1 s, the total time discrepancy would be approximately 3 or 4 min throughout the TKA surgery. Thus, in this study, TKA surgery on the left side (the surgeon’s non-dominant side) took longer than surgery on the right side because the surgeon was right-handed. The discrepancy in the median operative time between left- and right-sided surgery in this study was 4 min.

The Jonckheere–Terpstra trend test revealed differences according to the operative side in this study. It is supposed that the efforts to achieve effective instrumentation handoffs and deliveries were more remarkable on the non-dominant side, where there is much room for improvement (greater distance between the surgeon and the scrub nurse(s)), compared with the surgeon’s dominant side where there is less distance between the surgeon and the scrub nurse(s). This may explain the different findings on different operative sides.

There are three limitations in this study. First and foremost, the results in this study had limitations owing to the study’s retrospective nature based on using the medical record. Second, unlike a previous report that measured the time for each surgical step [10], the total operative duration was evaluated in this study. Third, the ratio of males to females was uneven. A female predominance of OA and TKA operations is a common finding of Asian patients [12,17,21,22,24,26]. Despite these limitations, the strength of this study was less surgeon-related bias than in previous studies. This was because all procedures, from skin incision to wound closure, were performed by a single experienced surgeon. Another strength is less impact of patient-related anatomical bias on operative time when comparing right and left knees, as surgery was performed in bilateral TKA patients.

## 5. Conclusions

Surgeons should recognize that large physique, especially obesity, when performing surgery on a surgeon’s non-dominant side in OA knee surgery may require longer operative times. This study proposes two suggestions to decrease operative time in this situation. One is that patients should maintain their appropriate BW before TKA. The second is that efforts should be devised to save time during instrumentation handoffs or deliveries between the surgeon and the nurse.

## Figures and Tables

**Figure 1 jcm-11-04867-f001:**
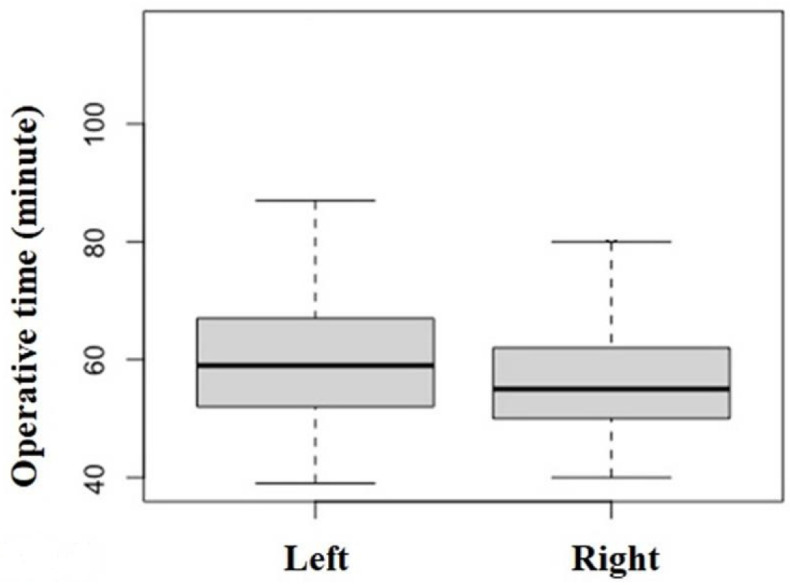
Comparison of the operative times for the left and right sides in total knee arthroplasty (TKA) (*p* = 0.015).

**Figure 2 jcm-11-04867-f002:**
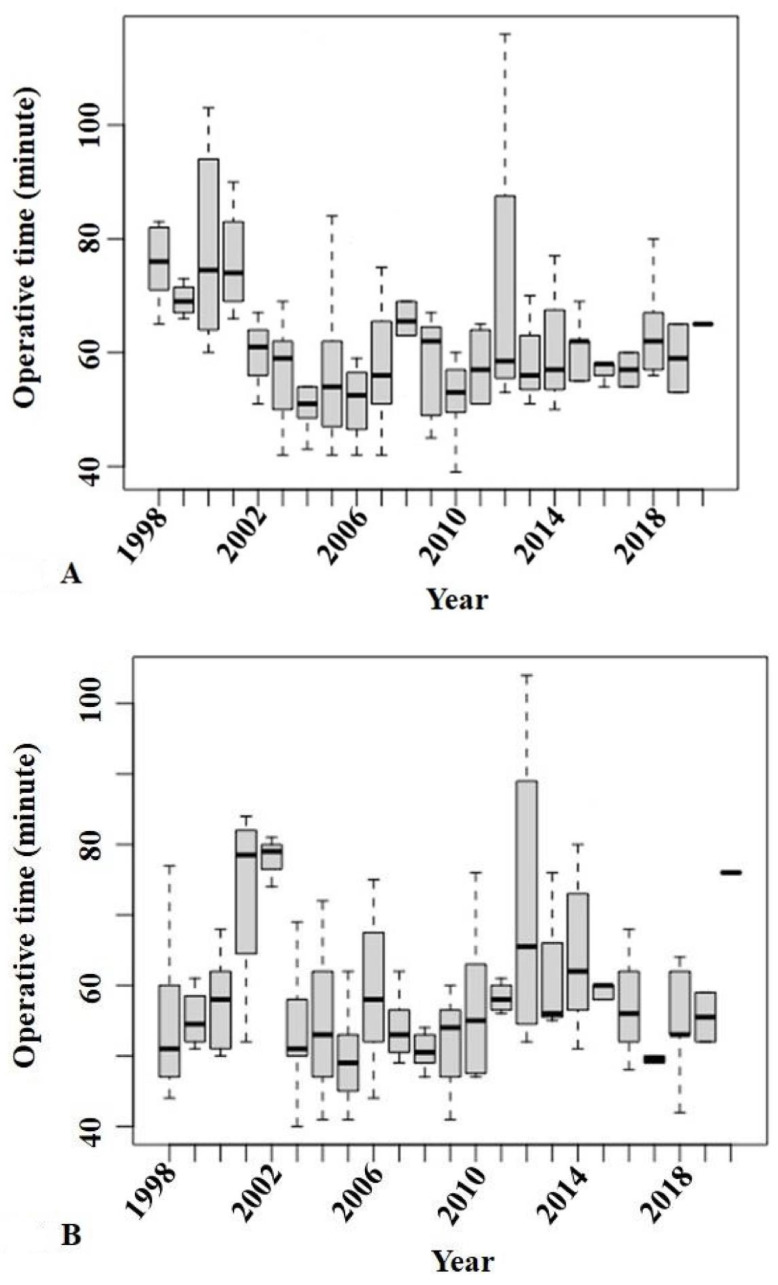
The Jonckheere–Terpstra test showed a significant trend in decreasing operative time with calendar year for surgeries on the left side (*p* = 0.037) (**A**) but not the right side (*p* = 0.795) (**B**).

**Figure 3 jcm-11-04867-f003:**
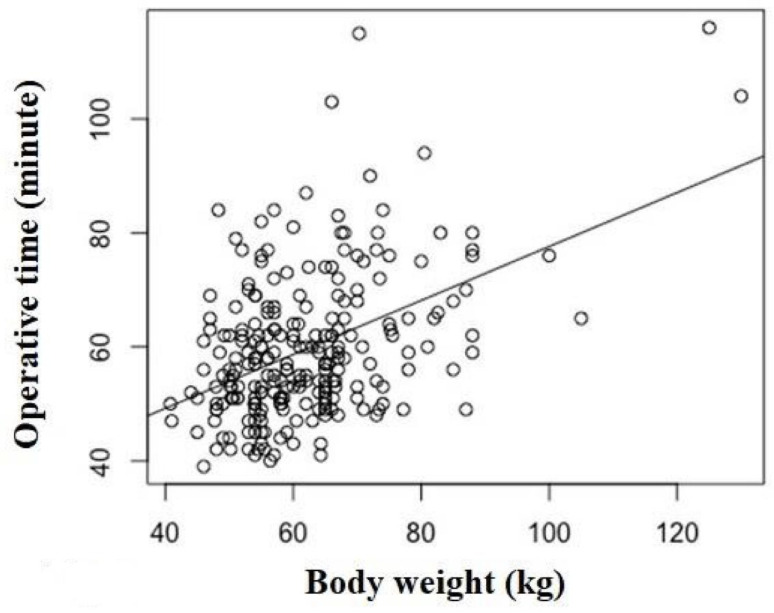
Correlation between operative time and body weight (BW) (r = 0.352, *p* < 0.001). BW showed a correlation with operative time in the multiple regression analysis.

**Table 1 jcm-11-04867-t001:** Patients’ background characteristics.

	Patients	Knees	
Gender M/F	15/104	30/208	
Side R/L	119/119	238/238	
ASA (I/II) [19]	6/113	12/226	
	**Median**	**Interquartile Range**	**Range**
Interval (days)	628	315, 1460	35–5110
Age (year)	73	69, 77	34–88
BH (cm)	151	146, 155	135–184
BW (kg)	59	54, 67	41–130
BMI (cm/kg^2^)	26	24, 29	19–42
HSS score [20]	45	37, 51	19–67
TFA (°)	181	179, 184	162–199

M, male; F, female; R, right; L, left; ASA, American Society of Anesthesiologists; BH: body height; BW, body weight; BMI, body mass index; HSS, hospital for special surgery; TFA, tibiofemoral angle.

**Table 2 jcm-11-04867-t002:** Results of the multiple regression analysis using a stepwise variable selection method.

	B	S.E.	β	Sig.	95%CI	
(Constant)	1025.183	255.308		0.000	522.188	1528.179
Body weight	0.505	0.058	0.488	<0.001	0.391	0.619
Calendar year of surgery	–0.495	0.127	–0.218	<0.001	–0.746	–0.245
Side of surgery (left; 0, right 1)	–3.815	1.407	–0.151	0.007	–6.586	–1.044

S.E., standard error; Sig., significance; CI, confidence interval.

**Table 3 jcm-11-04867-t003:** Correlations between operative time and the study variables by Spearman’s rank correlation coefficient.

Variable	r	P
Sex	0.188	0.004
Age (years)	–0.144	0.026
Body Height (cm)	**0.212**	**0.001**
Body Weight (kg)	**0.352**	**<0.001**
Body Mass index (kg/cm^2^)	**0.290**	**<0.001**
HSS score [20]	0.042	0.517
Tibiofemoral angle	0.090	0.167
ASA grade [19]	–0.112	0.084
Order (1st/2nd)	–0.120	0.065

Values in bold indicate statistically significant values. HSS score, hospital for special surgery score; ASA, American Society of Anesthesiologists.

## Data Availability

The datasets used and/or analyzed during the current study are available from the corresponding author on reasonable request.

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
