# Peer review of "Comparison of Operative Times in Primary Bilateral Total Knee Arthroplasty Performed by a Single Surgeon"

_jcm, 2022, doi:10.3390/jcm11164867_

Round 1

Reviewer 1 Report

This manuscript proposed a very nice cohort study treating the factors that may affect the operative time of total knee arthroplasty surgery. Compared with the published ones, the novelty of the presented output is the less technical and anatomical bias, which may favor a more solid and accurate conclusion on the factors that may affect the operative time. This kind of work is usually considered a difficult and time-consuming problem that depends simultaneously on several parameters, and the authors are commended for their efforts. Generally, the manuscript is well written and organized, and the method is appropriately designed; addressing the following comments would be beneficial.
1) It is beneficial for the interested readers to see a graphical conclusion linking the most important factors to the operative time.
2) How can some of those factors be controlled? It may be useful in the abstract.  
3) check the text again for some minor grammatical and typo errors.

Reviewer 2 Report

Dear Authors the topic is estremely interesting

As regards the introduction i suggest to improve this section introducing the complications of tka from aseptic mobilization and paifull knee prosthesis to knee malalignment citing the following articles:

Painful knee prosthesis: CT scan to assess patellar angle and implant malrotation

Spinarelli A. et al.

Muscles, Ligaments and Tendons JournalOpen AccessVolume 6, Issue 4, Pages 461 - 4661 October 2016

Baropodometry on patients after total knee arthroplasty

Notarnicola A. et al.

Musculoskeletal SurgeryVolume 102, Issue 2, Pages 129 - 1371 August 2018

As regards M&M, results and discussion, these sections are well described but it’s neccesary to add the complications due to prolonged operative time. In fact i suggest to improve the article by revealing the complications that you could reveal in order to underline the relationship between prolonged operative time and complications

Notwithstanding the limits of the study the conclusion is very important  
